# The Growth of *Vallisneria natans* and Its Epiphytic Biofilm in Simulated Nutrient-Rich Flowing Water

**Liman Ren [1], Yan Gao [2],*** , **Zhixin Hu [3], Xue Jiang [1] and Liuyan Yang [1]**

1  State Key Laboratory of Pollution Control and Resource Reuse, School of the Environment, Nanjing University, Nanjing 210023, China; renliman03@163.com (L.R.); eruditej@126.com (X.J.); yangly@nju.edu.cn (L.Y.)
2  School of Environment Science and Engineering, Nanjing Tech University, Nanjing 211816, China
3  School of Environmental Engineering, Nanjing Institute of Technology, Nanjing 211167, China; zxhu@njit.edu.cn
*  Correspondence: gaoyan@njtech.edu.cn; Tel.: +86-17751029507

**Abstract:** This paper investigates the effects of water flow on the growth and physiological indicators of the submerged macrophyte, *Vallisneria natans*, and the bacteria and algae community composition on its epiphytic biofilm-covered leaves. The authors set up a simulated flowing water laboratory experiment testing high nitrogen (N) and phosphorus (P) concentrations. Total chlorophyll and dissolved oxygen (DO) was significantly enhanced, and turbidity was reduced, thereby accelerating the growth of *V. natans*. These experiments were compared to another set of observations on a static group. The accumulation of malonaldehyde (MDA) in the dynamic groups was significantly higher than that in the static group. As an antioxidant stress response, the total superoxide dismutase (T-SOD) was also induced in plants exposed to nutrient-rich flowing water. The results of 16S rRNA high-throughput sequencing analyses showed that the water flow increased the bacteria community diversity of biofilm-producing bacteria with N and P removing bacteria, carbon cycle bacteria, and plant growth-promoting rhizobacteria on the epiphytic biofilm. This research determined that water flow alleviates the adverse effects of eutrophication when *V. natans* grows in water containing high N and P concentrations. Water flow also inhibits the growth of cyanobacteria (also referred to as blue-green algae) in epiphytic biofilm. The ecological factor of water flow, such as water disturbance and aeration measures, could alleviate the adverse effect of eutrophic water by providing a new way to restore submerged macrophytes, such as *V. natans*, in eutrophic water.

**Keywords:** water flow; *Vallisneria natans*; growth; epiphytic biofilm; bacteria; algae





## 1. Introduction

Submerged macrophytes can maintain the biodiversity and stability of an aquatic ecosystem. Their participation in the circulation of material and energy in water helps keep water clear, removes nutrient pollutants and reduces excessive algal growth, thereby successfully implementing water eutrophication remediation [1,2]. Recently, submerged macrophyte technology has attracted enormous interest in the restoration of non-eutrophic lakes and rivers by researchers and government departments seeking to improve water quality and aquatic ecosystems [3–5]. However, in eutrophic water, excess nutrients simultaneously promote the propagation of anaerobic bacteria and cyanobacteria as well as other eutrophic algae. These nutrients also cause a decrease in light and oxygen, which hinders the growth of submerged macrophytes [6] and produces anaerobic gases. One such gas is ammonia nitrogen which has toxic effects on plants [7]. Therefore, improving the content of dissolved oxygen (DO) and water transparency are essential to the restoration of submerged macrophytes.

Understanding submerged macrophytes and their epiphytic biofilm are key factors in water eutrophication remediation. Their specific and highly complex interactions interfere

with important ecosystem processes [8]. Research has recently found that the submerged macrophytes and their epiphytic biofilm are affected by a variety of abiotic factors, such as water movement, temperature, and nutrition [9–12]. Epiphytic biofilm is comprised of an autotrophic community made up of diatoms, green algae, and cyanobacteria, as well as a heterotrophic community consisting of bacteria, protozoa, fungi, and other microorganisms [13]. Research has revealed that plant morphological and physiological parameters are altered by water movement and the epiphytic biofilm microbial community structure changes in response to water flow; moreover, epiphytic biofilm activity increases during hydrodynamic transient storage [14–16]. Water flow promotes the activity of nitrifying microorganisms by increasing the concentration of DO, which is conducive to nitrification, thereby promoting the transformation of ammonia nitrogen [17]. An appropriate water flow can promote the exchange of oxygen and the supply of nutrients [18,19], thereby helping oxygen and nutrients reach the submerged microphytes in biofilm leaves where microorganisms affect the growth of epiphytic biofilm and the succession of the microbial community in submerged macrophytes [16,20,21]. However, it is a challenge for submerged macrophytes to exist in some heavily eutrophic aquatic ecosystems due to competition from anaerobic bacteria and cyanobacteria and other eutrophic algae, which is not conducive to the growth of submerged macrophytes [22]. How water flow changes the growth of submerged macrophyte and the community structure of bacteria and algae attached to the epiphytic biofilm in eutrophic water is essential to understanding and controlling eutrophication. Revealing this process is conducive to clarifying the impact of water flow on epiphytic biofilm-macrophyte interactions in eutrophic water.

Water flow has a great potential to improve the applications of epiphytic biofilm-macrophyte interactions in wastewater treatment with better management of the aquatic ecosystems under pressures of N and P pollution. A comprehensive study is needed to delineate the response of submerged macrophytes and their epiphytic biofilm in flowing water rich in nutrients, which can provide a new method to enhance the recovery of aquatic ecosystem restoration, such as water disturbance and aeration measures. In this study, *Vallisneria natans* was selected because it could effectively absorb N and P nutrients in water and inhibit algal bloom. Its growth indices of leaf length, leaf width, root length and wet weight, and the physiological and antioxidant defense indices of chlorophyll, soluble sugar, total protein, root activity, malonaldehyde (MDA) and total superoxide dismutase (T-SOD) were all investigated in microcosms to assess plant health in simulated flowing water rich in nutrients. Furthermore, the species and diversity of bacteria and algae dependent on the epiphytic biofilm of *V. natans* were all detected and analyzed, based on their responses to flowing water rich in nutrients. The purpose of this study is to explore the impact of water flow on phytoremediation in eutrophic water.

## 2. Materials and Methods

### 2.1. Pretreatment of V. natans and Sediments

The experimental *V. natans* seedlings were taken from Qianma Island in Jiangsu Province, China. Their mean height and root length were 26.35 ± 1.85 cm and 10.88 ± 1 cm, respectively, and the average number of leaves was 6. We cleaned the soil attached to the surface of leaves and roots with clean water; then, the seedlings were all transplanted to an indoor tank domesticated for about 20 days before use. Sediments for cultivating *V. natans* seedlings were collected by a Peterson dredger from the Jiuxiang River since it was used as a model eutrophic river (in the Qixia District of eastern Nanjing city). The seedlings were sealed, shaded and immediately transported back to the laboratory. After the preliminary removal of debris and stones, the sediment was dried at 30 °C then screened through 20 mesh sieves and fully stirred. The sediments were evenly spread in growth zone (II) of the experimental device, with a thickness of about 7 cm (Figure 1).

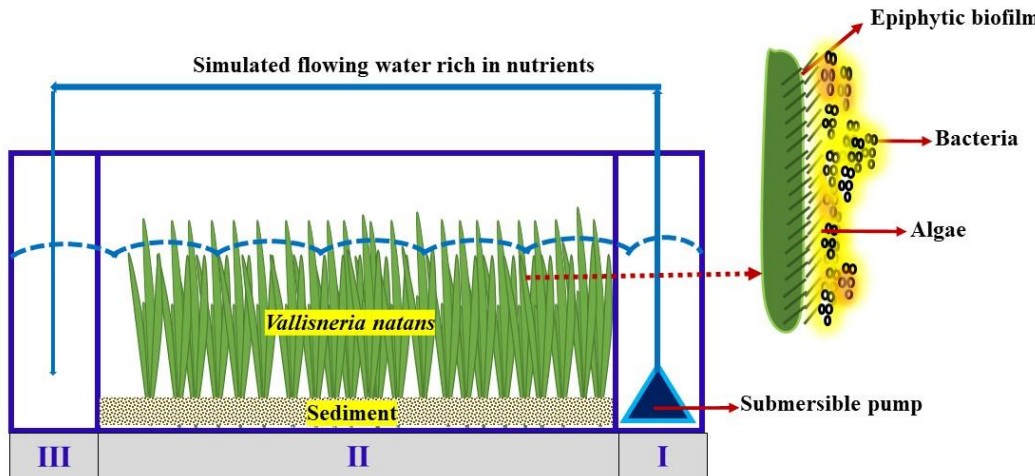

**Figure 1.** Experimental device and separate schematic of the *V. natans* epiphytic biofilm.

### 2.2. Experimental Setup and Experimental Conditions

Figure 1 shows the experimental setup of a polyvinyl chloride (PVC) plexiglass tank, with a length, width and height of 80 cm × 20 cm × 30 cm, respectively. It was divided into three parts. The 11 × 20 cm water distribution zone I has a JTP-5800 SUNSUN aquarium pump operating at 38 W from the bottom of the tank, separated from growth zone II by a perforated plate. The plate has four evenly distributed circular water holes with a diameter of 4 cm each. Growth zone II was 58 × 20 cm for at least 24 of the *V. natans* plants, which were evenly planted. Catchment zone III was the same size as zone I and was separated from the growing area by the perforated plate, while the submerged pump pipe fed into the catchment unit.

The simulated N and P nutrient-rich flowing water was prepared by dissolving ammonium chloride, sodium nitrate, potassium dihydrogen phosphate and sodium acetate in tap water to maintain concentrations of 12 mg·L$^{-1}$ of NH$_3$−N, 0.4 mg·L$^{-1}$ of NO$_3^-$-N, 0.2 mg·L$^{-1}$ of PO$_4^{3-}$-P, and 50 mg·L$^{-1}$ of chemical oxygen demand (COD) for the microcosm experiment. No microelements were added since they were available in the tap water. See Table S1 for amounts of NH$_3$-N, NO$_3^-$-N, TN, PO$_4^{3-}$-P, COD, pH and ORP in the water of growth zone II at the initial and end of the experiment. We put 48 L of N & P enriched water into the 21-cm deep tank. In dynamic groups, the flowing water rich in nutrients flows from the water distribution zone (II) to the growth zone (II) and then to the catchment zone (III) where the water is circulated at a water flow velocity of 35 cm·s$^{-1}$. There was no water flow in the static groups. Each sample has three sets of replicates. The illumination intensity was supplemented by an LED fluorescent lamp with an illumination intensity of 3000 lux. In the 12 h of light dark cycles, the temperature was maintained at about 24 °C to 26 °C The experimental period was 60 days under a no-water exchange policy. Water lost due to evaporation in the tanks were supplemented regularly to maintain the same water level. The DO was immediately measured by a portable Hach HQ30d multi-parameter analyzer, an LDO101-03 DO electrode (Hach Company, Loveland, CO, USA). The turbidity was measured by 2100Q HACH portable turbidimeter.

### 2.3. The Growth Indices of V. natans

The *V. natans* samples were collected on days 0, 20, 40 and 60 to reveal the effects of water flow on the growth as well as the physiological and antioxidant defense indices of *V. natans*. On each sampling day, at least three *V. natans* plants were selected from dynamic groups and static groups as to avoid sediment disturbance to the maximum extent. The length of the *V. natans* leaves and roots were measured with a ruler; the width of the leaves was measured with a Vernier caliper, and the fresh weight (FW) and dry weight (DW) of *V. natans* was measured with an electronic balance. The leaf powder was acquired by

grinding it thoroughly with liquid nitrogen, after freeze-drying by an FD-1C-80 freeze dryer (Beijing Boyikang Experimental Instrument Co., Ltd., Beijing, China), then put it into a −80 °C refrigerator for storage.

*2.4. The Physiological and Antioxidant Defense Indices of V. natans*

2.4.1. Chlorophyll

The chlorophyll concentration was measured by using ultraviolet (UV) spectrophotometry according to the method described by Sartory and Grobbelaar (1984) with minor modifications [23]. After cutting and mixing the leaves of *V. natans*, about 0.5 grams of fresh leaves were put in a 15 mL Corning centrifugal tube. Ten mL of 95% ethanol was then added to the tube, which was shaken prior to placement in the dark for 24 h. We shook the mixture to make the chlorophyll extraction complete. After the *V. natans* leaves turned completely white, the extraction solution was measured at 665 nm and 649 nm wavelengths by a Shimadzu UV-1800 UV-Vis spectrophotometer (Shimadzu Scientific Instruments, Tokyo, Japan) to obtain the concentrations of chlorophyll according to [23]:

$$C_a = 13.95 \times OD_{665} - 6.88 \times OD_{649} \tag{1}$$

$$C_b = 24.96 \times OD_{649} - 7.32 \times OD_{665} \tag{2}$$

$$\text{Chlorophyll (mg·g}^{-1}) = (C_{(a+b)} \times V)/FW \tag{3}$$

where $C_a$ represents the concentration of the $C_a$ chlorophyll (mg·L$^{-1}$); $OD_{665}$ represents the measured $OD_{665}$ value of the sample; $OD_{649}$ represents the measured $OD_{649}$ value of the sample. The $C_b$ chlorophyll (mg·L$^{-1}$) represents the concentration of chlorophyll *b*; V presents the extracting solution (L), and FW presents the fresh weight of the *V. natans* (g) leaves.

2.4.2. Total Protein

The determination of total protein concentration in the leaves of *V. natans* was measured by the Coomassie brilliant blue method [24], which included weighing about 0.5 g of leaf powder (DW). We added saline solution (the mass fraction of NaCl was 0.9%) according to the ratio of leaf weight (g). Thus, the volume of normal saline solution (mL) is 1:9. The mixture was then homogenized mechanically under ice bath conditions; then, the homogenate was centrifuged at 4 °C at 2500 r·min$^{-1}$ for 10 min, taking the supernatant as a standby. We then added 50 μL of supernatant, 50 μL of the protein standard sample, and 50 μL of ultrapure water into a Corning centrifuge tube. At this point, 3 mL of Coomassie brilliant blue chromogenic solution was added to set the coloring for 10 min. The absorbance value of each tube was determined at 595 nm by an ultraviolet and visible light from a Shimadzu V1800 UV-Vis spectrophotometer (Shimadzu Scientific Instruments, Tokyo, Japan) after the corresponding standard pretreatment and reagent addition were completed. The samples' protein concentrations were calculated as [24]:

$$P \text{ (gprot·L}^{-1}) = [(OD_v - OD_b)/(OD_{sv} - OD_{sb})] \times S \times D \tag{4}$$

where P represents the concentration of total protein (gprot·L$^{-1}$); $OD_v$ represents a sample's measured $OD_{595}$ value; $OD_b$ represents the $OD_{595}$ value of the blank; $OD_{sv}$ represents the measured $OD_{595}$ value of the standard; $OD_{sb}$ represents the $OD_{595}$ value of a blank; *S* represents the standard concentration, and D represents the sample dilution rate.

2.4.3. Root Activity

The metabolic root activity was determined by estimating the reduction amount of triphenyltetrazolium chloride (TTC) [25]. The *V. natans* roots were washed (FW) with deionized water and dried with paper towels, then weighed immediately, the roots were then put into a 25 mL beaker with an added 5 mL of 0.4% 2, 3, 5-triphenyltetrazolium chloride (TTC) solution, 5 mL of 0.1 mol·L$^{-1}$, and phosphate buffer solution (pH 7.5). All

the contents were thoroughly mixed and the solution was incubated at 37 °C for 2 h in the dark. The final step was to add 2 mL of 1 mol·L$^{-1}$ H$_2$SO$_4$ solution to terminate the reaction.

We took out the *V. natans* roots and dried them with filter paper. We then put the roots into another beaker and completely immersed them in 10 mL methanol before transferring the root solution to a constant temperature incubator at 37 °C until the apical segment turned white completely. The absorbance was measured at 485 nm with methanol as the reference, and the reduction amount of tetrazole in the system was calculated according to the standard curve expressed as [25]:

$$\text{Root activity } (\mu g \cdot (g \cdot h)^{-1}) = (c \times m)/(w \times h) \tag{5}$$

where c represents the reduction amount of tetrazole calculated by the standard curve (μg). m represents the dilution ratio of the extract. w represents the root weight (g), and h represents the coloring time (h).

### 2.4.4. Soluble Sugar, Malondialdehyde and Total Superoxide Dismutase

The soluble sugar concentration of *V. natans* leaves (DW) was measured by the anthrone colorimetry method, using a plant soluble sugar detection kit (Nanjing Jiancheng Bioengineering Institute, Nanjing, China). The content of soluble sugar mainly included soluble monosaccharides, oligosaccharides and polysaccharides with a calculated value based on sample protein concentration. The determination of MDA concentration in the *V. natans* leaves (DW) was measured by the microplate method using a plant malondialdehyde kit (Nanjing Jiancheng Bioengineering Institute). T-SOD activity was determined with the assay kit according to the manufacturer's protocol (Nanjing Jiancheng Bioengineering Institute, China).

### 2.5. The Species and Diversity of Bacteria and Algae Dependent on the Epiphytic Biofilm of V. natans

At the end of the experiment, we initially washed the mud off the *V. natans* leaves and then placed them in a white polyethylene basin. A clean bristle brush was used to gently brush the surface attachment of *V. natans* leaves (about 10 g) into 2 L of sterilized ultrapure water. One L deionized water was filtered with a polycarbonate membrane with a pore size of 0.22 μm. The membranes were then collected and stored at −80 °C for 16S rDNA bacterial analysis through high-throughput sequencing (Sangon Biotech (Shanghai) Co., Ltd., Shanghai, China, Contract No was 16S183980). The remaining 1 L water sample was fixed with Lugol's iodine solution and formaldehyde solution, which was then left settled for 48 h in a pear-shaped glass separating funnel. The solution was also concentrated at 50 mL for algae identification and then stored in darkness before examination by a Nikon Eclipse E200 photomicroscope (Nikon Instruments (Shanghai) Co., Ltd., Shanghai, China).

### 2.6. Data Analysis

The Origin ver. 9.0 (OriginLab, Northampton, MA, USA, 2015) and SPSS ver. 23.0 software (IBM Corp., Endicott, NY, USA, 2015) were used for mapping and data (Mean ± SD) statistics. The differences between the different data groups were analyzed by one-way ANOVA. A *p* value of less than 0.05 could be interpreted to declare that the differences were statistically significant.

## 3. Results and Discussion

### 3.1. Effects of Water Flow on the Growth of V. natans

The growth of *V. natans* was in good condition in the dynamic group as the water flow rate was 35 cm·s$^{-1}$, and the growth of leaves and roots significantly increased compared with the static group. After 20 days of the *V. natans* early planting stage, the leaf length of *V. natans* increased by 51.45% in the nutrient-enhanced simulated flowing water. The mean leaf length increased to 74.5 ± 7.0 cm with the leaf elongation reaching 192% after 60 days of continuous cultivation in the dynamic group ($p < 0.05$). However, in the static

group, the mean leaf length was only $31.3 \pm 7.0$ cm with a leaf elongation of only 15.4% ($p < 0.05$) compared with the leaves in the dynamic groups; moreover, the rot became visible. This may have been caused by insufficient DO and the toxic effect of ammonium ions at the initial stage. Notably, the leaf width lessened under the water flow condition. The root length was also longer in the dynamic group ($10.8 \pm 1.0$ cm) than in the static group ($9.4 \pm 0.9$ cm) ($p < 0.05$) (Figure 2a–d and Table S2), thereby revealing that the water flow promoted the growth of both the *V. natans'* leaves and roots.

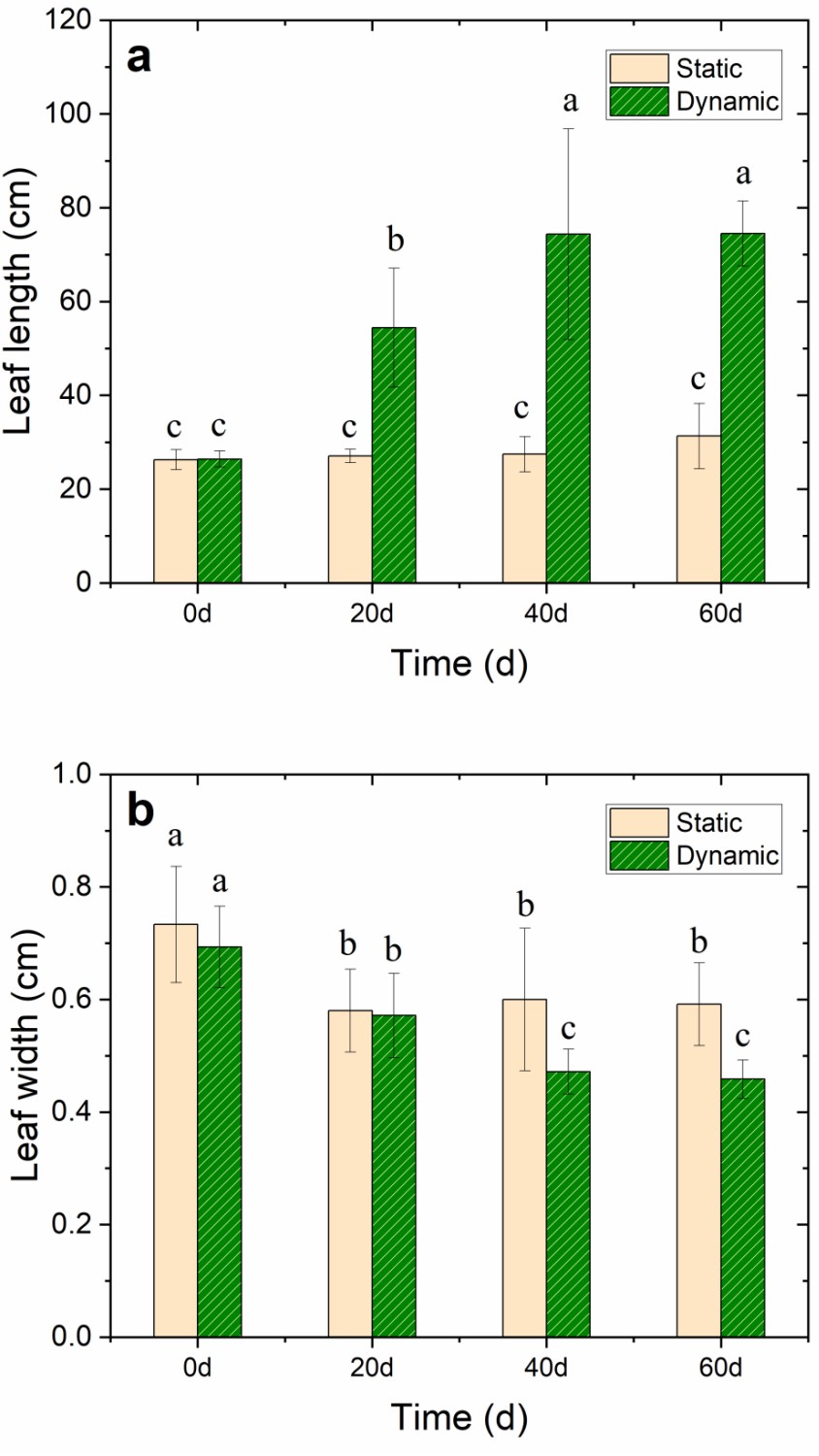

**Figure 2.** *Cont.*

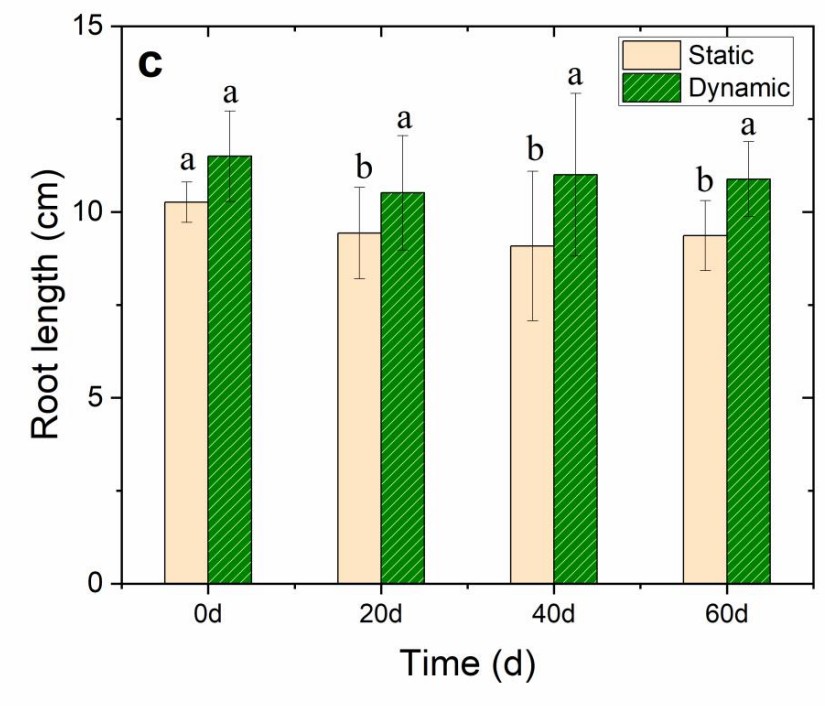

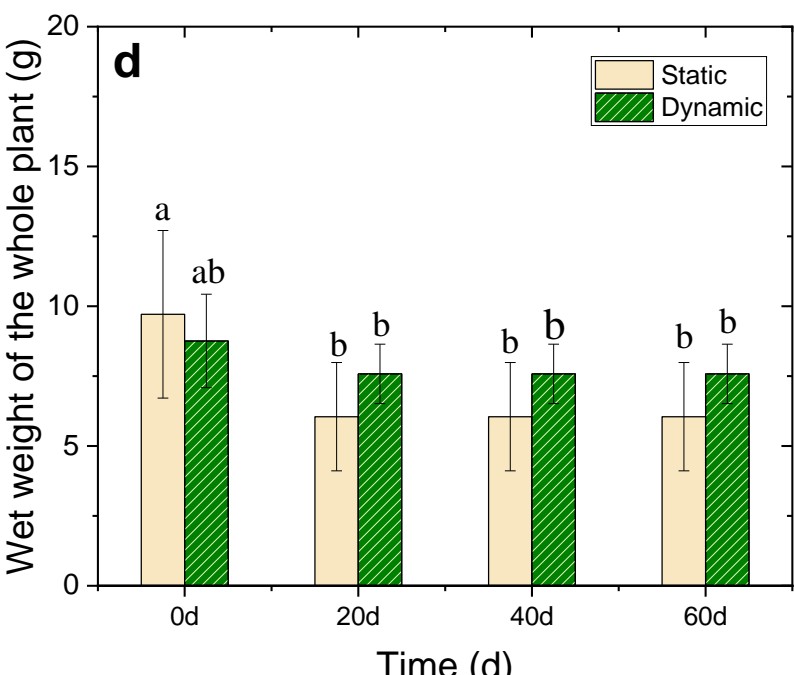

**Figure 2.** Effects of 35 cm·s$^{-1}$ water flow velocity on (**a**) leaf length, (**b**) leaf width, (**c**) root length, and (**d**) wet weight of the whole *V. natans* plant during the 60-day cultivation period. The lowercase letters above each ivory static or green dynamic bar indicate significant differences ($p < 0.05$) in response to different treatments, and all data had a mean value of ±SD.

DO and turbidity are the main environmental factors affected the growth of *V. natans*, so naturally, these factors were detected during the cultivation period. The results (as shown in Figure 3a,b) verify that the water flow increased the DO in water as the DO concentration was significantly higher than that of the static group ($p < 0.05$) which increased to 7–8 mg·L$^{-1}$ on day 60. As in this static group, the DO concentration was about 0.12 mg·L$^{-1}$ in the static group initially; then, the DO concentration was maintained at 2–3

mg·L$^{-1}$ from day 10 to day 60, which was not conducive to the growth of *V. natans*. As for turbidity, the water flow kept the turbidity value of the water body at about 2–3 NTU, which improved as the water transparency; furthermore, as the improved light reached the bottom, the growth of *V. natans* increased. The turbidity of the static group fluctuated greatly. Meanwhile, the highest turbidity reached 20 NTU, which was significantly higher than the dynamic group ($p < 0.05$) and was unfavorable to the *V. natans* growth rate.

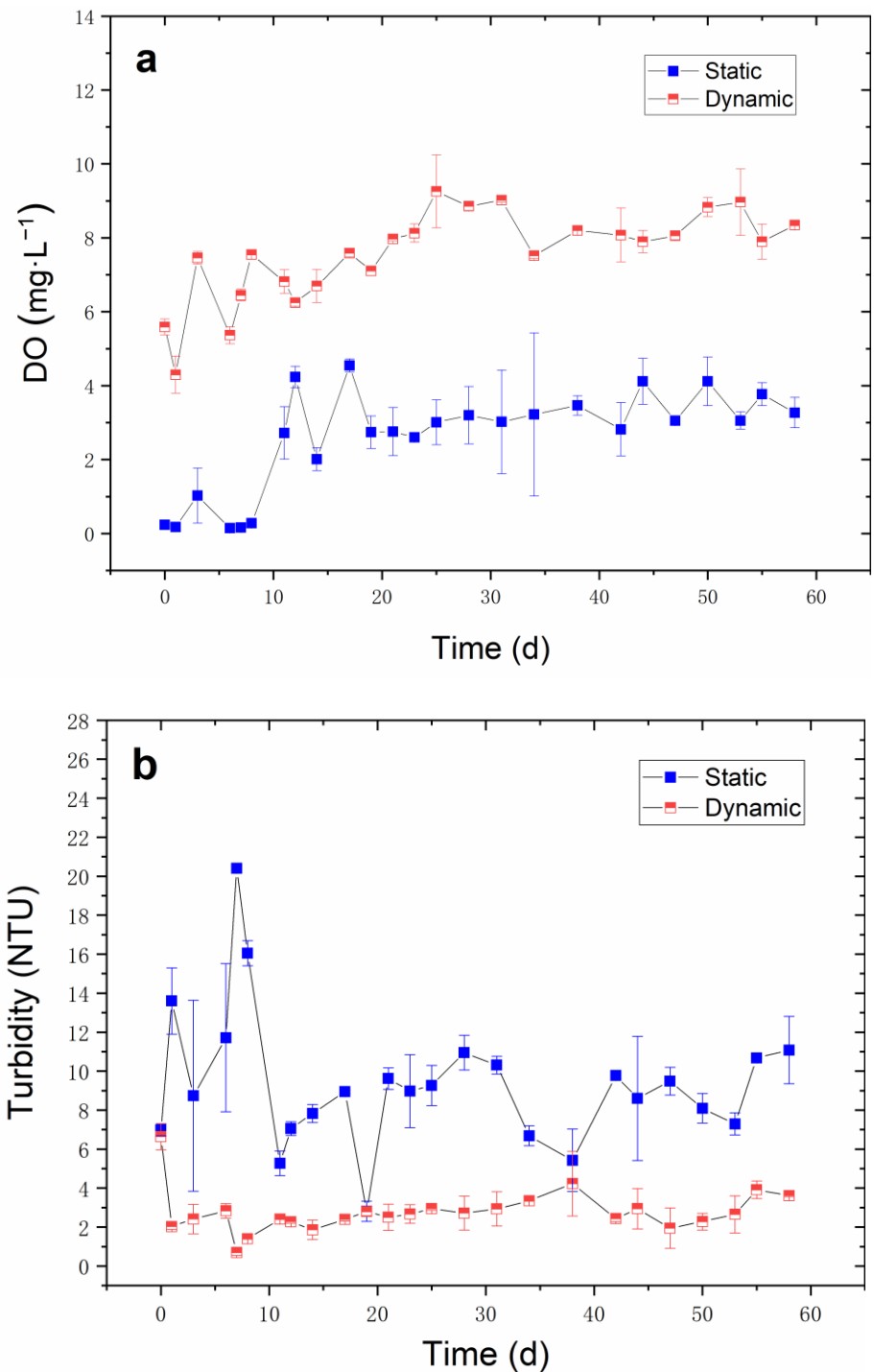

**Figure 3.** Effects of water flow velocity (35 cm·s$^{-1}$) on nutrient-rich simulated water. (**a**) DO (mg·L$^{-1}$) and (**b**) turbidity (NTU) during the 60-day cultivation period. All data had a mean value of ±SD.

Extensive research has been carried out on the natural water flow in lakes and other waterbodies, which enhances the growth of aquatic plants. Nilsson [26] observed positive flow effects (up to 30 m·s$^{-1}$) on the macrophyte cover and the number of species present. Crossley et al. [27] found that under the condition of flowing water, the leaf, total dry weight, and leaf area of aquatic plants increased by 29%, 58% and 40%, respectively. This type of activity affects the absorption of nutrients and/or soluble inorganic carbon by *V. natans* leaves, thereby promoting the growth of *V. natans* [18]. Water flow significantly increased the DO concentration in water and alleviated the inhibition of low DO on the growth of *V. natans*. Studies have shown that the toxicity of non-ionic ammonia will increase with the decrease of DO concentration [28]. Fan et al. [29] found that $NH_3$ concentration in water was inversely proportional to the DO concentration which directly affects the activity of ammonia-oxidizing microorganisms meant to mediate the transformation of $NH_3$. Thus, higher DO concentrations could reduce the ammonia toxicity to aquatic plants and aquatic animals [2]. Due to the decrease of DO, the growth of anaerobic microorganisms and algae led to the decrease of water transparency in the static group (Figure 3b). The transparency and water depth play a fundamental role in the growth and reproduction of submerged macrophytes [30]. Water flow enhanced the DO concentration, and the toxic effect of the anaerobic environment on *V. natans* was also reduced. However, excessive water flow may be detrimental to the growth. Therefore, the growth of *V. natans* requires appropriate water flow since water flow promotes the increase of DO and reduces turbidity in water, thereby promoting the growth of *V. natans*.

### 3.2. Effects of Water Flow on the Physiological and Antioxidant Defense Indices of V. natans

The chlorophyll concentration in the leaves of *V. natans* can directly affect photosynthetic intensity. Light drives the photosynthesis, producing carbohydrates that allow plants to grow [31]. In this study, the total chlorophyll concentration in the leaves of *V. natans* in dynamic groups increased significantly ($p < 0.05$), from the initial concentration of $0.6 \pm 0.04$ mg·g$^{-1}$ (FW) to $1.1 \pm 0.1$ mg·g$^{-1}$ (FW) on the 60th day (Figure 4a and Table S3). Previous studies have found that under the lower water velocity, the photosynthesis rate of submerged macrophytes increased with the increase of water velocity [32]. Water flow improved the water body transparency since water turbidity was reduced (Figure 3b), which increased the light intensity under the water surface and promoted the chlorophyll concentration of *V. natans*, thereby promoting photosynthesis.

Soluble sugar concentration in plants most often indicates the carbon nutrition of plants, which is an important intermediate product of photosynthesis, respiration and carbohydrate storage of plants. Carbon nutrition in plants and the subsequent benefits play an important role in the C-N metabolism of plants [33–35]. On the other hand, under the stress of high salinity and hypoxia, plants accumulate a certain amount of osmotic adjustment substances, such as soluble sugar and proteins, to improve the concentration of cell fluid and adaptation to the stress environment. The concentration of soluble sugar in the water flow group reached the maximum on the 20th day; furthermore, during the entire experiment, the protein concentration was higher than the static group without one significant difference (Figure 4b). As the cultivation time increased, the total protein concentration of *V. natans* in dynamic groups was a little higher than that in the static group (Figure 4c and Table S3). The role of proteins includes constructing new cells, tissues, and organs as well as sustaining life activities [16].

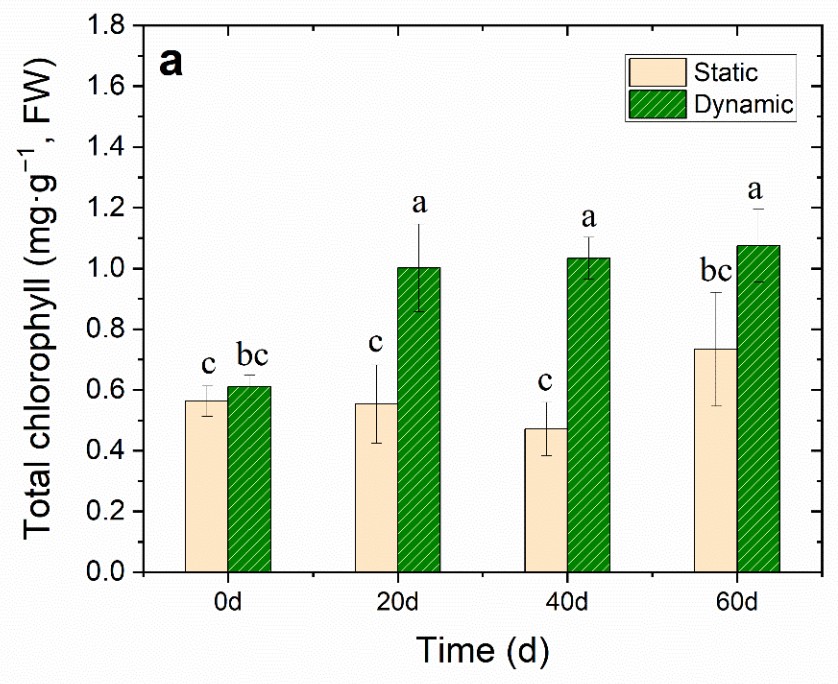

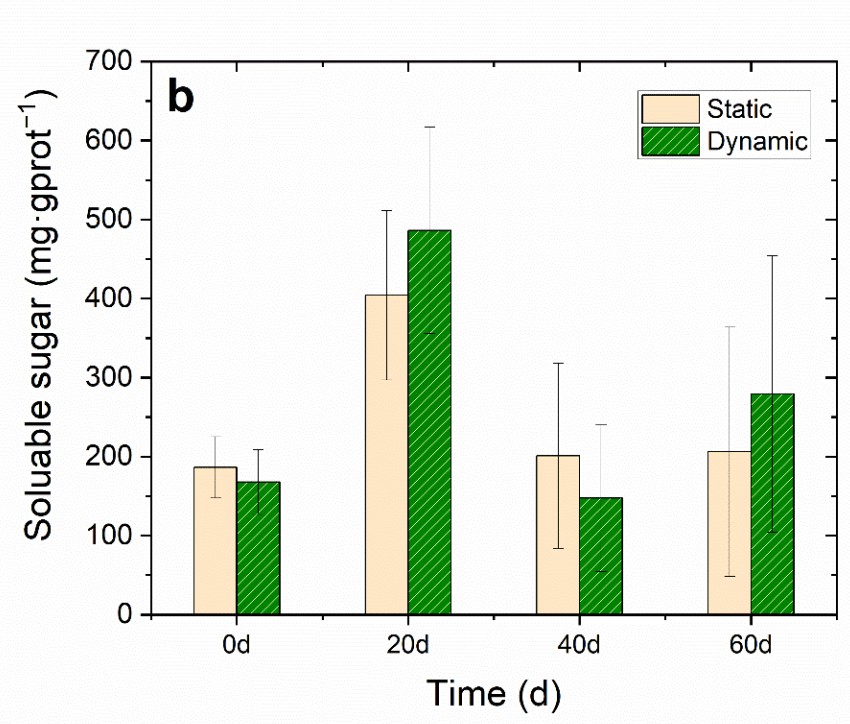

**Figure 4.** *Cont.*

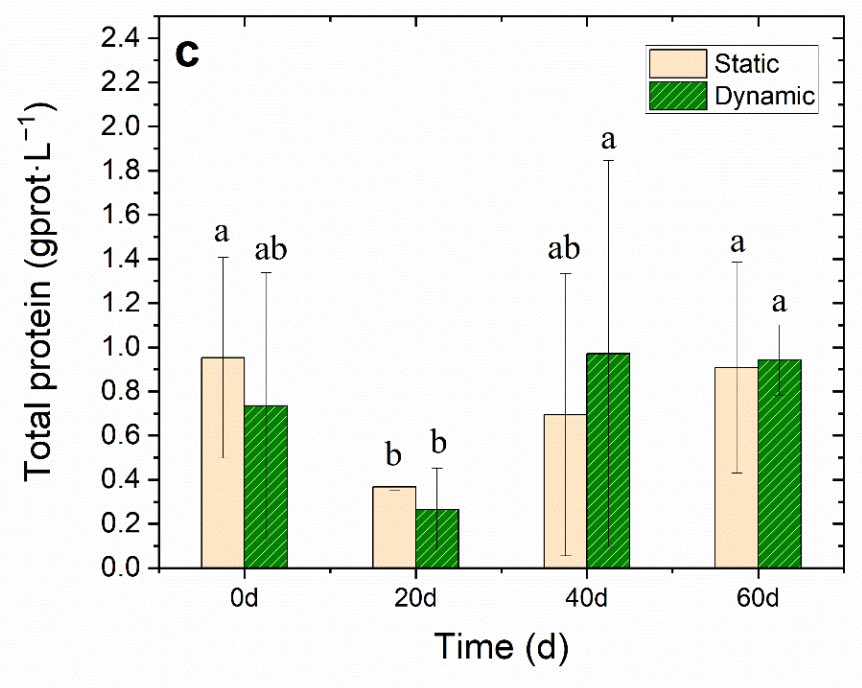

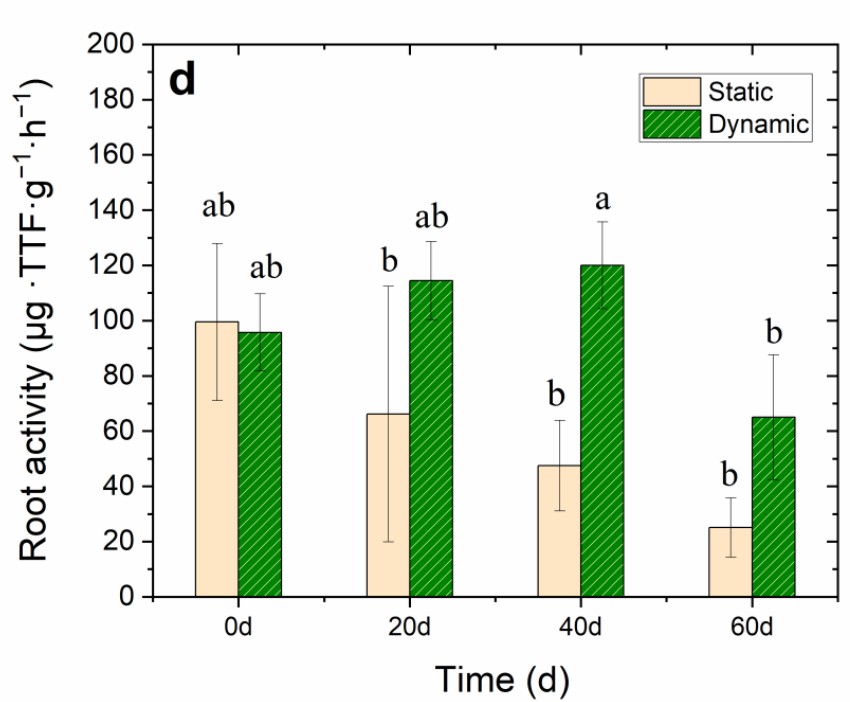

**Figure 4.** *Cont.*

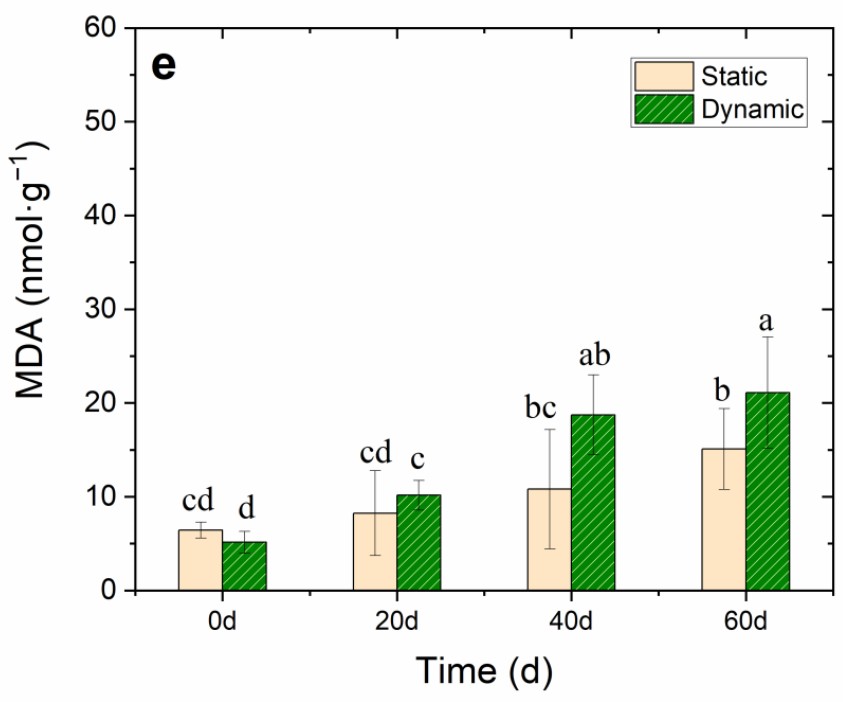

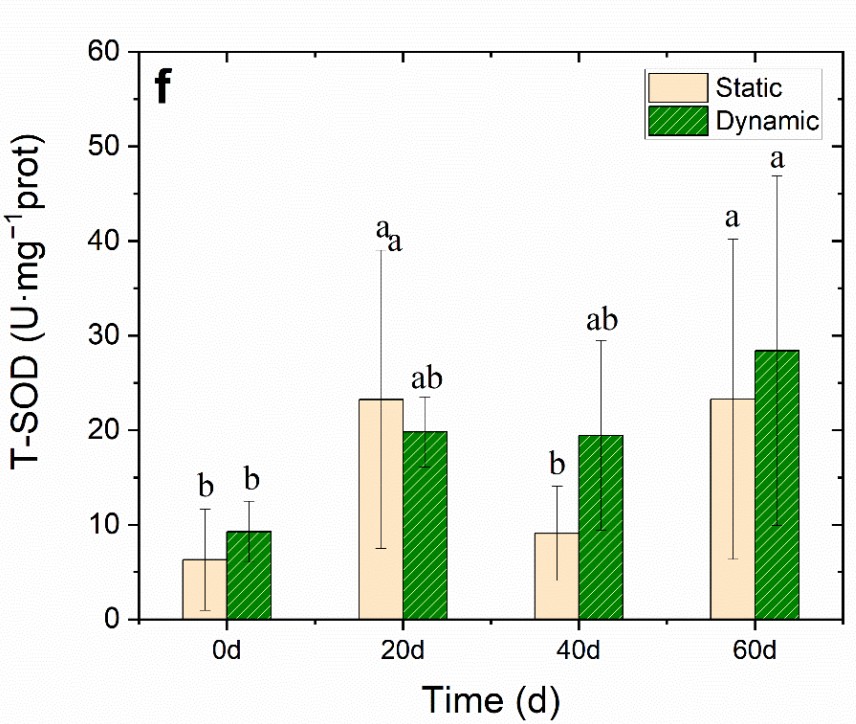

**Figure 4.** Effects of water flow on (**a**) total chlorophyll (mg·g$^{-1}$, FW), (**b**) soluble sugar (mg·g$^{-1}$ prot), (**c**) total protein (mgprot·g$^{-1}$, DW), (**d**) root activity (μg·g$^{-1}$·h$^{-1}$, FW), (**e**) MDA (nmol·g$^{-1}$, DW) and (**f**) T-SOD activity (U·mg$^{-1}$ prot) of *V. natans* during the 60-day cultivation period. All data had a mean value ± SD. The lowercase letters above the ivory static bars and green dynamic bars indicate significant differences ($p < 0.05$) among different treatments.

The root system of submerged macrophytes not only fixes the plant, but also provides nutrition. Root activity is one of the important indices needed to estimate the growth status of submerged macrophytes. The growth status and activity level of roots directly affect the

growth status, nutritional status and yield level of submerged macrophyte [32]. The results showed that after the cultivation period of 40 days, the root system activity of the dynamic groups maintained a high level—as high as $120 \pm 15.7$ μg·g$^{-1}$·h$^{-1}$, FW—compared to the control static group, which decreased to $47.5 \pm 16.4$ μg·g$^{-1}$·h$^{-1}$, FW—a significant decrease ($p < 0.05$) of about 74.8% lower than that of the initial level (Figure 4d). The results showed that the DO concentration in water increased to 7–8 mg·L$^{-1}$ on day 60, which was a significant difference when compared to the static group ($p < 0.05$). Therefore, the root system of *V. natans* was able to carry out normal aerobic respiration and increase its root activity. When the plant was anoxic, the energy metabolism in the root was disordered. The aerobic respiration of the plant roots slowed down or stopped, and the energy supply of the root cells was lacking. At the same time, the low oxygen condition caused by the lack of DO concentration hindered the absorption of nutrients in sediments containing *V. natans* roots [36]. Thus, water flow increased the DO concentration in water, thereby promoting the *V. natans* root activity.

MDA accumulation usually indicates that serious peroxidation damage is in the plant membrane. When the MDA accumulates in plants, it interferes with photosynthesis, respiration, and other metabolism activities in plant cells. Test results showed no accumulation of MDA in the first 20 days, However, MDA accumulated rapidly from the 20th day to the 40th day. On day 60, the accumulation of MDA in the dynamic group was significantly higher ($21.1 \pm 5.9$ nmol·g$^{-1}$, DW) than that in the static group ($15.1 \pm 4.3$ nmol·g$^{-1}$, DW) (Figure 4e and Table S3), which may have been caused by the stress of water flow. The concentration of T-SOD in *V. natans* increased ($28.4 \pm 18.5$ U·mg$^{-1}$ protein) to decrease the free radicals in the leaf, which reduced the MDA concentration. T-SOD is an important protective enzyme of the enzymatic defense system, which can maintain the dynamic balance between the production and elimination of reactive oxygen species, eliminate the continuous production of free radicals in the body, protect the membrane structure, avoid the damage of reactive oxygen species to plants, and make plants slow down or resist stress to a certain extent [16]. Therefore, water flow stress may lead to the increase of reactive oxygen species in plants, and an increase in the activity of the antioxidant defense system of the T-SOD enzyme, which can quickly eliminate any harmful reactive oxygen species.

### 3.3. Water Flow Effects on Bacteria Biodiversity of V. natans Epiphytic Biofilm

At the end of the experiment, 16S rDNA of bacteria living on the epiphytic biofilm of *V. natans* were analyzed to further explore the water flow effects on their biodiversity and functions. As test results show, after 60 days of cultivation, when compared to the static group, the Shannon value of the bacterial community attached to the epiphytic biofilm leaves of *V. natans* increased in dynamic groups, while the Simpson value decreased (Table S4). Furthermore, the number of operational taxonomic units (OTUs) also showed an upward trend. The change trend of the Chao1 index and ACE index were consistent with the change trend in the OTU number. The results indicated that water flow increased the species and biodiversity of bacteria living on the V. *natans* epiphytic biofilm. Through the analysis of the bacterial composition, the dominant bacterial genera in dynamic groups were *Pseudomonas* (8.6%), *Methylophilus* (7.7%), *Acinetobacter* (4.8%), *Exiguobacterium* (4.64%), *Sphingomonas* (4.5%), *Gemmobacter* (3.5%), *Actinobacter* (3.5%) and *Massilia* (3.4%), in addition to *Hydrogenophaga*, *Novosphingobium*, *Brevundimonas*, *Methylobacterium* and *Noviherbaspirillum*. The proportion of *Pseudomonas* and *Methylphilus* in the static group was only 1.6% and 2.9%, respectively (Figure 5). Through the analysis of the above bacterial species and functions, the authors found that water flow increases bacteria community diversity such as biofilm-producing bacteria, N and P removal, carbon cycle and plant growth-promotion of rhizobacteria on the epiphytic biofilm of *V. natans*, thereby promoting nutrient circulation and protecting *V. natans* growth.

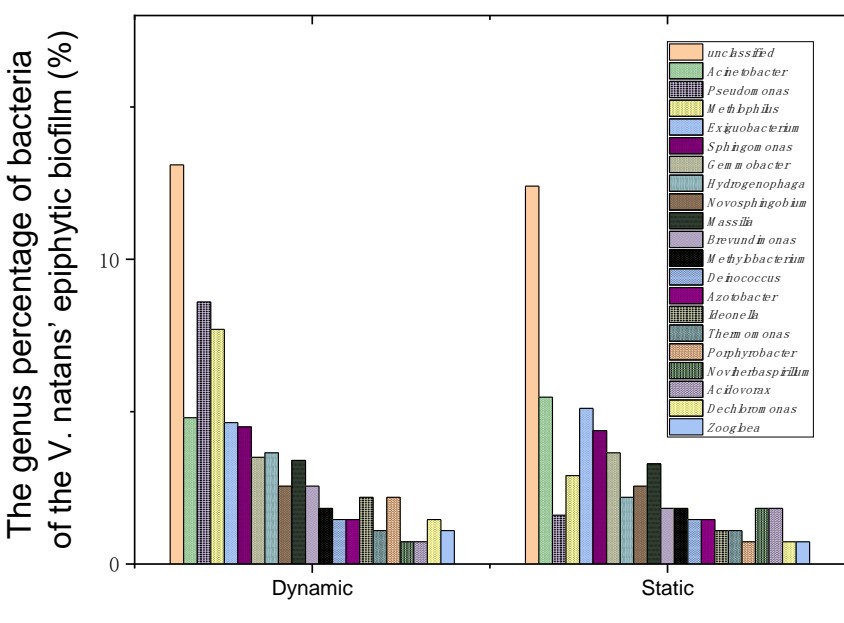

**Figure 5.** Effects of water flowrate of 35 cm·s$^{-1}$ on the bacteria of the *V. natans*' epiphytic biofilm after 60 days of cultivation.

Submerged macrophytes provide a habitat for microorganisms, so that microorganisms can attach to the surface of submerged macrophyte leaves and the biofilm microbial community structure can change in response to water flow in eutrophic water [16,37]. Among these attached microorganisms, bacteria have the most species and the largest number, and their functions are also more diversified [38]. The bacteria of *Pseudomonas* can improve plant nutrition, produce antibiotics and plant growth regulators, degrade toxic substances, improve plant microenvironment and promote other biological control effects. It was found that *Pseudomonas* bacteria can play an active role in the growth of plants [39]. In addition, *Pseudomonas* and *Brevundimonas* are also important biofilm-producing bacteria [40]. The rhizospheric *Pseudomonas* can significantly promote N removal and biofilm formation [41]. The genera *Pseudomonas*, *Novosphingobium*, and *Hydrogenophaga* are all capable of removing N and P [39,42]. *Methylophilus* and *Methylobacterium* can utilize methanol, glucose and fructose. Their abundant presence on the surface of the biofilms-leaves of *V. natans* in a dynamic group plays a key role in global carbon cycle [30]. Meanwhile, some bacteria candidates are used as plant growth-promoting rhizobacteria based on their abilities of salt stress alleviation, production of indole acetic acid and acetoin, and siderophore [43]. *Exiguobacterium* spp can promote rhizosphere growth and decompose organic pollution [44]. Thus, the water flow increases the bacteria community diversity of the *V. natans* epiphytic biofilm conducive the increased DO concentration, decreased turbidity and the good growth of *V. natans*.

### 3.4. The Algae Living in the Epiphytic Biofilm of V. natans

After 60 days cultivation, the algae living in the *V. natans* epiphytic biofilm were collected, and the species were analyzed. The results showed 17 species in two phyla, including 15 species of Bacillariophyta and two species of Cyanophyta. Bacillariophyta and Cyanophyta were the main groups of algae in the *V. natans* epiphytic biofilm in the dynamic groups with water flow throughout the experiment described in Section 2.2 (Figure 6a). In the static group with no water flow, four phyla encompassed 37 species, with 25 species belonging to Bacillariophyta. Five species belong to Cyanophyta; one species belongs to Euglenophyta, and six species belong to Chlorophyta. Bacillariophyta was the dominant

alga phylum living on the *V. natans* epiphytic biofilm because of its easy access, accounting for 88% in dynamic groups and 68% in control static groups.

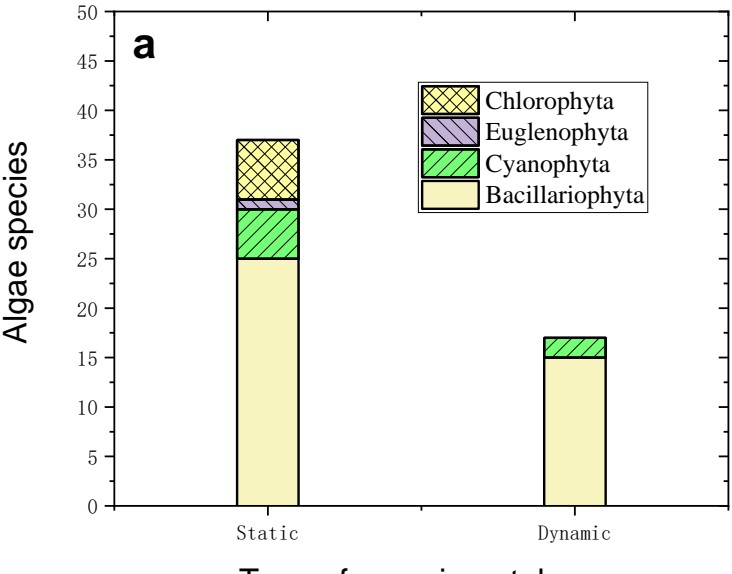

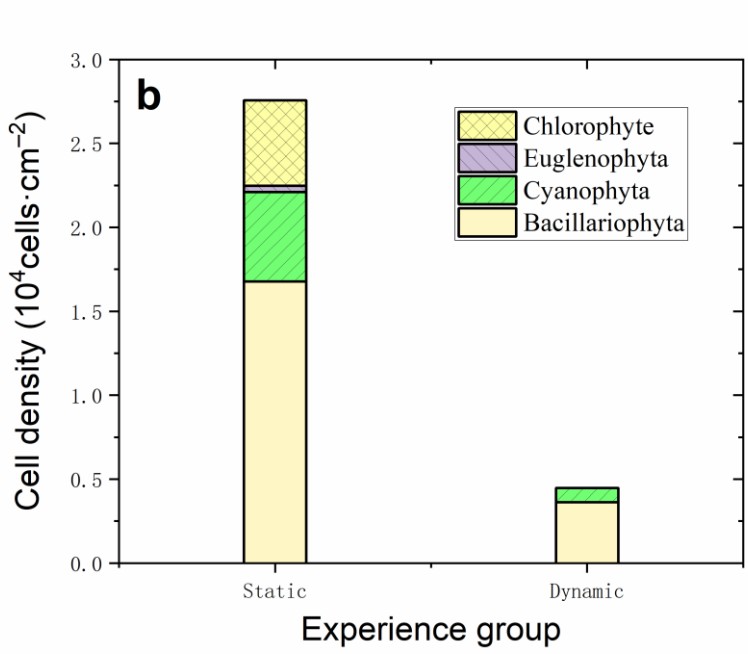

**Figure 6.** Effects of water flow on (**a**) algae species and (**b**) algae cell density on the epiphytic biofilm of *V. natans* after 60-day cultivation.

Further data analysis showed the total abundance of algae living in the epiphytic biofilm of *V. natans* under a water flow, i.e., the dynamic group, was $0.448 \times 10^4$ cells·cm$^{-2}$. The abundance of Bacillariophyta and Cyanophyta was $0.362 \times 10^4$ cells·cm$^{-2}$ and $0.086 \times 10^4$ cells·cm$^{-2}$, respectively. In the static group, the total abundance of algae in the epiphytic *V. natans* was as high as $2.758 \times 10^4$ cells·cm$^{-2}$. The abundance of Bacillariophyta, Cyanophyta and Chlorophyta was $1.677 \times 10^4$ cells·cm$^{-2}$, $0.534 \times 10^4$ cells·cm$^{-2}$, and $0.509 \times 10^4$ cells·cm$^{-2}$, respectively. Therefore, water flow reduced the abundance of algae in the epiphytic biofilm of *V. natans*, especially eutrophic algae (Figure 6).

The diversity indices of algae living on the *V. natans* epiphytic biofilm is shown in Table S5. The results showed that the Shannon-Wiener index and the Margalef index of the algae living on the epiphytic biofilms in dynamic groups were lower than those of the static group, and the Pielou index was greater than that of the static group. Thus, water flow reduced the diversity of algae living in the epiphytic biofilm of *V. natans*; moreover, the community structure tended to be simple.

The dominant species of algae living on the *V. natans* epiphytic biofilm are shown in Table 1. There were eight dominant species in dynamic groups, and the main dominant species were *Fragillaria.* sp.1, *Cocconeis placentula*, *Aphanizomenon* sp, and *Nitzschia* sp.1. Due to the quantity of species (13 dominant species are in the static groups), and the number of dominant species (*M. varians*, *Aphanizomenon* sp, *Scenedesmus Bijuga*, and *Fragillaria.* sp.2) also increased the species and dominance of algae in the static groups.

**Table 1.** Effects of water flow on the dominant genus, species and dominant index on the epiphytic biofilm of *V. natans*.

| Phylum | Genera | Dominant Species | The Dominant Indices | |
| --- | --- | --- | --- | --- |
| | | | Dynamic | Static |
| Bacillariophyta | *Cocconeis* | *C. placentula* | 0.125 | 0.036 |
| | *Melosira* | *M. varians* | 0.058 | 0.113 |
| | *Fragillaria* | *F.* sp.1 | 0.029 | 0.072 |
| | | *F.* sp.2 | 0.135 | 0.032 |
| | *Nitzschia* | *N.* sp. | 0.096 | 0.036 |
| | *Navicula* | *N.gracilis* | / | 0.068 |
| | | *N.* sp. | / | 0.045 |
| | *Gomphonema* | *G.* sp. | / | 0.027 |
| | *Achnanthes* | *A.* sp. | 0.038 | / |
| Cyanophyta | *Aphanizomenon* | *A.* sp. | 0.115 | 0.090 |
| | *Oscillatoria* | *O.* sp. | 0.077 | 0.034 |
| Chlorophyta | *Scenedesmus* | *S. bijuga* | / | 0.077 |
| | *Characium* | *C.* sp. | / | 0.059 |
| | *Chlamydomonas* | *C.* sp. | / | 0.023 |

In a natural aquatic ecosystem, a micro-interface system (such as the *V. natans* leaf-epiphytic biofilm interface) develops on the leaves surface, which is composed of organic matter, algae and microorganisms [45]. The massive growth of algae within the macrophyte's biofilm community can contribute to the decline of submerged macrophytes. There is a competitive relationship between submerged macrophytes and their resident algae, especially for the competition of nutrients and light. Other organisms within biofilm can also hinder the absorption of inorganic carbon by host plants thereby affecting the growth of submerged macrophytes [19]. Studies have shown that with the increase of algae biomass within biofilm, the fresh weight (FW) of *V. natans* decreases greatly, and with the increase of biomass from the biofilm algae, the chlorophyll concentration and photosynthesis rate of submerged macrophytes also decrease significantly [46]. When the submerged macrophytes are in the state of decay or death, they will reduce or no longer secrete allelochemicals, which weakens the algal inhibition effect of submerged macrophytes, causing large quantities of algae residing in the *V. natans* biofilm to grow especially in the eutrophic algae of Bacillariophyta and Cyanophyta. In conclusion, water flow can promote the growth of *V. natans* by reducing the number of algae in the epiphytic biofilm of *V. natans*, and at the same time, the promotion of water flow on the growth of *V. natans* will further improve the algae inhibition ability.

## 4. Conclusions

From the microcosm experiment involving the dynamic and static simulated water rich in nutrients for cultivation of *V. natans*, it was determined that the water flow (35 cm·s$^{-1}$) significantly increased the DO concentration and reduced the turbidity of nutrient-rich water. At the end of the experiment, the leaf length and root length increased significantly, with the growth rate reaching 192% ($p < 0.05$) with high total chlorophyll (1.1 ± 0.1 mg·L$^{-1}$, FW), higher total protein (9.4 ± 0.2 mg prot·g$^{-1}$, DW), and soluble sugar (279.2 ± 174.8 mg·g$^{-1}$ prot) than the static group since the photosynthesis intensity of the leaves increased during the water flow testing. However, water flow also led to some stress, and the antioxidant content of the T-SOD increased from 9.3 ± 3.2 to 28.4 ± 18.5 U·mg$^{-1}$ prot, while the MDA concentrations and free radical content in the leaves decreased. From the investigation of bacteria on the *V. natans* epiphytic biofilm under water flow, the diversity and abundance of bacteria increased. Compared with the static group, the biofilm-producing bacteria, N and P removal, carbon cycle and plant growth-promotion of rhizobacteria such as *Pseudomonas* and *Methylophilus* in the attached bacteria, would all have a positive effect on the growth of *V. natans*, thereby promoting nutrient circulation and protecting epiphytic biofilm growth. At the same time, the diversity of the algae inside the biofilm decreased, the inhibition effect of this algae on the growth of *V. natans* also decreased. However, static simulated water rich in N and P nutrients caused algae to grow, especially the eutrophic algae of Bacillariophyta and Cyanophyta. Water flow maintained a healthy bacteria and algae community on the surface of *V. natans* epiphytic biofilm. This study provides theoretical guidance for the recovery of submerged plants in a eutrophic water environment by promoting water flow measures such as aeration.

**Supplementary Materials:** The following are available online at https://www.mdpi.com/article/10.3390/w14142236/s1, Table S1: Nitrogen, phosphorus, COD, pH and ORP in the water of growth zone II at the beginning and end of the experiment, Table S2: The growth indices of *V. natans* at the beginning and end of experiment, Table S3: The physiological and antioxidant defense indices of *V. natans* at the beginning and end of experiment, Table S4: Statistic results for table of bacteria diversity indices living on *V. natans* periphytic biofilm, Table S5: Diversity index of algae on the *V. natans* epiphytic biofilm.

**Author Contributions:** Conceptualization, L.R. and L.Y.; methodology, L.R., Z.H., X.J. and Y.G.; software, Y.G. and L.R.; validation, Y.G.; formal analysis, Y.G. and L.Y.; investigation, Y.G. and L.Y.; resources, L.R.; writing—original draft preparation, Y.G.; writing—review and editing, Y.G.; visualization, Y.G.; supervision, L.Y.; project administration, L.Y.; funding acquisition, Y.G. and L.Y. All authors have read and agreed to the published version of the manuscript.

**Funding:** This research was funded by the Jiangsu Provincial Natural Science Foundation of China (BK20190320), the National Natural Science Foundation of China (51908277) and the National Water Pollution Control and Treatment Science and Technology Major Project (2017ZX07204002).

**Institutional Review Board Statement:** Not applicable.

**Informed Consent Statement:** Not applicable.

**Data Availability Statement:** Not applicable.

**Acknowledgments:** This work was supported by the State Key Laboratory of Pollution Control and Resource Reuse of Nanjing University and funded by the Jiangsu Provincial Natural Science Foundation of China (BK20190320), the National Natural Science Foundation of China (51908277) and the National Water Pollution Control and Treatment Science and Technology Major Project (2017ZX07204002).

**Conflicts of Interest:** There are no conflict of interest to declare.

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
