# Peer review of "The Growth of Vallisneria natans and Its Epiphytic Biofilm in Simulated Nutrient-Rich Flowing Water"

_water, doi:10.3390/w14142236_

Round 1
Reviewer 1 Report
A microcosm growth experiment was conducted on Vallisneria natans in a facility enabling to use a controlled circulating water flow. Growth traits and ecophysiological and biochemical parameters of plants together with the properties of the epiphytic biofilms on the plant surface were investigated in this experiment in relation to water eutrophication and plant tolerance of this factor. I suppose that this study and combination of all these study factors is very beneficial and meritorious and falls within the scope of Water. However, I have found also many shortcomings of this study and have tried to list them below. Unfortunately, several flaws are considerable and deteriorate the value of this study. They are usually associated with the Methods.
The English of the paper is sometimes clumsy and evidently contains a lot of mistakes; a thorough English correction by a native speaker is necessary.
My specific comments (page/line):
Abstract: l. 14-15: The water flow was conducive to the growth of the leaves (74.5 ± 6.95 14 cm) and roots (10.8 ± 1.01 cm) of V. natans, simultaneously promoted the accumulation of root activity (64.96 ± 22.63 μg TTF·(g·h) −1)..... the sentence does not give a sense: it is normal that leaves and roots grow, but what what the growth under the flow? Increased or reduced or what?? Rephrase this sentence. What is TTF? Define! Again, what is MDA?? Explain!
p.1, l.23: N and P removal bacteria?? May be better: N and P removing bacteria?
p.1, l.24: rhizobacteria bacteria?? better: rhizobacteria (only);
p.1, l.40: correctly: successfully used;
p.2: define all abbreviations again;
p.2, l.56: [15], : correctly: [15].
p.2, l.63-65: Whether the waterflow will improve the adaptability of the submerged macrophytes and an investigation of submerged macrophytes and the affiliated microbes, and algae as evidenced in epiphytic biofilm is required to better understand the process of phytoremediation in eutrophic water. : this sentence does not give a sense and should be rephrased.
p.2, l.71: correctly: water.
p.2, l.81: „Their mean height and root length were 26. 35 ± 85 cm“ : is it correct such a great variance? Otherwise, the authors should better round up all numerical values throughout this study, in this case 26.4 cm.
p.3: could you state the pH of the water between the star and end of the experiment? 3000 lux light intensity is not so much for Vallisneria. Which light source? Could you express the light in other units, too (quanta or energy)? Moreover, the authors do not state that they exchanged the water. Thus, what changes occurred in the water composition during the 60 days of the growth experiment?
p.3, l.121 and throughout the paper: not „indexes“, but correctly „indices“,
p.3, l.126: I understand that FW was measured as the fresh material could then be used for biochemistry. Yet, have the authors estimated the DW/FW ratio = Dry Matter Content in a minimum sample for overcalculation? FW is not so exactly estimated due to variable blotting of the material. The same question holds also for roots used for biochemical parameters.
p.4, l.144: in the literature, chlorophyll is almost always expressed per DW, not per FW; DMC would be very useful in this respect.
p.4, l.147: „we weighed about 0.5 g of fresh leaf powder“.... : if it was fresh leaf, then it could not be powder. Better: „fresh leaf biomass“ or „fresh leaves“;
p.4, l.149: what is „saline“?? Maybe „saline solution“ or so? I suppose that this expression is unclear here; correct.
p.4, l.159: (gprot·L−1): = gram of protein per litre; well, but what does it represent in plant/leaf biomass?? Nothing! This expression evidently gives the concentration of proteins per volume in the determination but the authors have not overcalculated this for leaf or plant biomass! They should express the content of proteins in the unit: mg protein.g-1 FW(or DW) and this unit should be used in all figures.
p.4, l.162: “S represents the concentration of standard concentration;“ : correctly: „S represents the standard concentration;“
Generally for all methods used, literature citations should be stated thoroughly illustrating the respective methods!
p.5, l.180-181: explain which sugars are included in this estimation – it is very important.
p.5, l.189 and on: „After the experiment, initially washed the mud off the roots and leaves of V. natans and then placed them in a white polyethylene basin. Use a clean bristle brush to gently brush the surface attachment of V. natans into 2 L sterilized ultrapure water. 1 L deionized water was filtrated......“ The English here is not suitable for a paper – rephrase. Spell: „One L“; correctly: „filtered“.
p.5, l.204: two-way ANOVA: what are the experimental factors? Flowing and ...??? Time? Explain. What is the n for one sampling? State this.
Figure 2 and other relevant figures: what do the variation bars represent: SE or SD? Explain.
p.5, l.217, Fig. 2d: The authors did not comment that the FW of their plants even in the dynamic variant decreased significantly during the first 20 days and were constant afterwards for the next 40 days! It means that even in the dynamic variant, the plants did not grow but only enlarged leaves and roots. Is it normal??? Why the plants did not increase their biomass? It seems that something was wrong if the plants were not able to grow and increase the biomass during the experiment. The authors should thoroughly explain this weak point. Otherwise, the authors should use the term of „leaf elongation“ rather than „growth rate“, which is usually reserved for increase in biomass and should be used for changes in biomass like Fig. 2d.
Fig. 4b: not „soluable sugar“ but „soluble sugar“ or „sugars“;
p.15, l.375: „Bacillariophyta and 2 species of Cyanophyta. Diatoms and Cyanobacteria.... : unite the terminology: Bacillariophyta or diatoms and Cyanophyta or Cyanobacteria?
Fig. 6 a,b: correctly: Chlorophyta; Experience group or Experimental group???
References: some references have non-abbreviated journal names; correct thoroughly.
In conclusion, the study bears interesting datasets but the fact that the experimental V. natans plants did not grow during the experiment (Fig. 2d), shows that something in the experiment was wrong and the authors should explain this – they have neglected and overlooked this fact. Also for numerous mistakes, I would recommend a major revision and a thorough English correction.
Reviewer 2 Report
Comments and Suggestions for Authors
The manuscript by Ren et al reports “Effects of Water Flow on the Growth of Vallisneria Natans and Epiphytic Biofilm in Highly Eutrophic Water. The effects of water flow on the growth and physiological indicators of the submerged macrophyte, Vallisneria natans, and the bacteria and algae community on its epiphytic biofilm-covered leaves were comprehensively investigated in eutrophic water with high nitrogen (N) and phosphorus (P) concentration. I recommend a major revision of the manuscript. Some concerns and suggestions are listed below for the author's attention.
- The introduction part needs to be enhanced by adding and discussing recent references. The authors can follow the recent references for improving the introduction part.
- Write a brief description in the introduction about the definition Vallisneria natans, how is biofilm formed? And Biofilm development stages.
- The equations of protein concentration of sample, the concentrations of chlorophyll a and b and Root activity, need to references.
- What is growth indexes of leaf length, leaf width, root length and wet weight, and the physiological and antioxidant defense indexes of chlorophyll, soluble sugar, total protein, root activity, MDA and T-SOD , at zero time and sixty days later (Explain theses in a table).
- What is the role of Vallisneria natans on phytoremediation in eutrophic?
- Please, improve the discussion section with some numerical data and previous studies.
- What were the characteristics of the impact of water used flow on phytoremediation in eutrophic?
- What were the microbial properties of biofilms as TEM analysis and SEM images of microbes?
- Provide a comparison table that investigates in detail the Vallisneria natans on phytoremediation of various bacteria or algae adsorbents.
- The novelty of the work should be emphasized in the last paragraph of the introduction. There are lots of similar works in literature. It should be mentioned which gap this work fills in the literature.
- The conclusion part must be enriched with all the valuable data of the research findings.
Author Response
"Please see the attachment."

Reviewer 3 Report
Dear Authours
Please add information which aspects of the presented article are new to restore submerged plants in eutrophic conditions of aquatic ecosystems.
Best regards
Author Response
"Please see the attachment."

Author Response
"Please see the attachment."

Round 2
Reviewer 1 Report
I have been given the revised manuscript on Vallisneria natans and biofilms for review. Although many mistakes have been corrected and the quality of the paper has greatly improved I have found many mistakes to be corrected (page/line, see below). I have not seen any Cover Letter explaining how the authors have corrected the mistakes (or explained why they have NOT corrected them). Unfortunately, several evident mistakes were reported in my first review but the authors have ignored them entirely:
p.1, l.17: ”the accumulation of root activity (64.96 ± 22.63 μg·(g·h) −1 )“: this expression does not give any sense – what is it??? Explain! The figures should be rounded up to only one decimal point. What type of variability is shown – SD or SE?
p.2, l.50: better: „concentration of dissolved oxygen“;
p.2, l.54: better: „Research has recently found ......“;
p.2, l.74: „is required to studied“: correctly: „is required to study“ or „to be studied“;
p.2, l.83: correctly: „because it could....“;
p.3, l.128: „LED fluorescent lamp“: it is probably non-sense as LED lamp cannot be fluorescent: correct!
p.3, l.130-133: „The experimental period was 60 days under no water exchange, and the water loss due to evaporation in the devices were supplemented regularly. Which was selected considering the aeration effect and growth of submerged plants in the process of water restoration.“ : the latter sentence needs rephrasing.
Which water was added – tap water or distilled water? In my first review, I requested to add data on pH of the water during the experiment (between the start and the end) but the authors have strictly ignored this request. If the water has NOT been replaced during the experiment, how the water composition was changing in the aquaria with 24 plants during the 60 days of the experiment? Again, no response of the authors! I suppose that water chemistry even at the start of the experiment is very poorly explained as the authors added some known nutrients to the base line of tap water of unknown composition! The authors should state what was the initial water composition +pH and what were the changes of the composition during (or at the end) the experiment. Otherwise, such an experiment has a very low value!
p.3, l.133: „There was no water flow in the static control groups.“: this sentence is twice and should be deleted.
p.4, l.164: However, chl. content is usually expressed per DW, not FW. Therefore, the authors should state the content per DW or at least to state the factor Dry Matter Content for recalculation for DW.
p.5, l.168: What is „normal saline solution“??? It is quite unclear. If 1 M NaCl, then state its composition directly.
p.5, l.184: perhaps better: „The metabolic root activity.....“;
p.5, l.188: „5 mL of 0.1 mol·L−1 , pH 7.5 phosphate buffer solution,“: it shoudl be better word ordered: e.g.: „5 mL of 0.1 mol·L−1 phosphate buffer solution (pH 7.5), .......“
p.6, l. 211-213: These sentences should be rephrased for correct English.
p.6, l.226: You should state here whether the shown variance bars are SD or SE throughout the manuscript but it should be stated for each figure legend!
p.6, l.231-232: „After 20 days of early planting stage of V. natans as the biomass growth was not obvious.“ : this sentence does not give a sence – correct!
p.6, l.237 and Fig. 2: „and the rot became visible.“ Does it mean that in the static variant, the leaves started rotting? It would deserve a greater attention. Moreover, as shown in Fig. 2, although the leaves of both variant more or less elongated, their growth (on FW level) was more or less negative (see Fig. 2d)!!! It is an important issue but the authors have neglected and ignored this negative fact although I did thoroughly ask for its explanation. Therefore, the authors cannot speak anymore on plant growth as it was only negative! The static plants might suffered from hypoxia but the dynamic plants had enough oxygen. Nevertheless, the authors added 12 mg ammonium-N per litre which may be killing concentration especially at (mildly) higher pH. The authors have nowhere stated the pH values of the solutions but if the pH of the dynamic variant was even only 8.00, there might be a toxic concentration of NH3; in the case of the static variant, the pH might be even much higher and, thus, the impact of the toxic [NH3] also much higher. On the basis of this disorder, it is very difficult to evaluate the results of this experiment when even the „better“ variant did not grow. I am sorry but I have serious doubts whether this paper should be published with these data on plant „growth“ but the authors have quietly ignored this issue!
Figure 2 and other relevant figures: what do the variation bars represent: SE or SD? Explain.
p.6, l.243: „water flow increased as the DO concentration“ : correctly: „water flow increased DO as the DO concentration......“;
p.6, l.251-252: „which was significantly lower than the dynamic group (p < 0.05), which was unfavorable to the growth of V. natans.“ The sentence is in controversion with Fig. 3b: change it e.g. to: „which was significantly higher than the dynamic group (p < 0.05) and was unfavorable to the growth of V. natans.“
p.7, l.269: correctly: „Water flow enhanced.....“;
p.7, l.270: „and thus promoted the growth of V. natans,“ : this is not true!!! As it follows from Fig. 2d, there was only negative growth also in the dynamic variant. The water flow only mitigated the damage to static plants but in itself, it did not lead to any positive growth!
p.11, l.308-309: ”Most of the enzymes needed by plants in the process of metabolism belong to proteins, and“ : This sentence is very futile as ALL enzymes are proteins; delete it.
Fig. 4b: units and legend: „Effects of water flow on (a) total chlorophyll (mg·g −1 , FW), (b) Soluble sugar (mg·g −1prot, DW), (c) total protein (gprot·L−1 , DW), (d) root activity (μg·g −1·h−1, FW), (e) MDA (nmol·g −1, DW) and 351 (f) T-SOD activity (U·mg−1 prot, DW): these units in the graphs and legends b, c and f are wrong:
b: „Soluble sugar (mg·gprot-1, DW)“: this unit is non-sense. Sugar content is usually expressed in % DW or in mg/g DW. It could be expressed per gram of proteins but why DW??? Correct!
c: total protein: should be expressed in mg prot. per g DW; the stated unit shows the protein concentration in a solution during the analytical procedure;
f: should be correctly in: U.mg-1 prot., but why DW??? Correct these wrong units!
Fig. 6: The explanation of the 4 groups is sufficient only once, in a or b.
p.16, l.404: „one specie belongs“: „species“;
p.18, l.459: „the leaf length and root length increased significantly, with the growth rate reaching 192%“ : this is wrong!!! Only leaf length increased but as it follows from Fig. 2c, root length did slightly decreased between the start and the end of the experiment!!! Leaf elongation is NOT growth rate!!! There was negative growth rate (i.e. FW decrease) of both variants (see Fig. 2d), which the authors cannot admit.
p.18, l.460-464: the units are wrong similarly to those in Fig. 4; see above for the explanation.
References: There are still minor formal mistakes in journal names: e.g. Water Res. (no full stop after Water as it is not abbreviated).
In conclusion, although the manuscript greatly improved its quality many important flaws have remained non-corrected though they were commented in my previous review. The main flaws are too high ammonium concentration in the water, absence of pH and water chemistry measurements during the experiment for 60 d, thus, a possibility of ammonia toxicity but without any possibility to verify it, negative growth rate even in the dynamic variant. I have great and serious doubts on the correctness of the experiment. Formal mistakes can be corrected but experimental mistakes cannot be overcome. Therefore, I cannot accept the results and their interpretation. Moreover, there are many important shortcomings in the paper and English needs a correction. I suggest to reject the paper.
Reviewer 2 Report
Accepted after language revision.
